# Structural Protein Effects Underpinning Cognitive Developmental Delay of the *PURA* p.Phe233del Mutation Modelled by Artificial Intelligence and the Hybrid Quantum Mechanics–Molecular Mechanics Framework

**DOI:** 10.3390/brainsci12070871

**Published:** 2022-06-30

**Authors:** Juan Javier López-Rivera, Luna Rodríguez-Salazar, Alejandro Soto-Ospina, Carlos Estrada-Serrato, David Serrano, Henry Mauricio Chaparro-Solano, Olga Londoño, Paula A. Rueda, Geraldine Ardila, Andrés Villegas-Lanau, Marcela Godoy-Corredor, Mauricio Cuartas, Jorge I. Vélez, Oscar M. Vidal, Mario A. Isaza-Ruget, Mauricio Arcos-Burgos

**Affiliations:** 1INPAC Research Group, Fundación Universitaria Sanitas, Bogotá 111321, Colombia; misaza@keralty.com; 2Grupo de Genética Médica, Clínica Universitaria Colombia y Clínica Pediátrica Colsanitas, Bogotá 111321, Colombia; carestrada@keralty.co (C.E.-S.); davidserranomsc@gmail.com (D.S.); henry.chaparro@urosario.edu.co (H.M.C.-S.); olgalondono0821@gmail.com (O.L.); 3Grupo de Bioinformática, Laboratorio de Clínica Colsanitas, Bogotá 110221, Colombia; lunrodriguez@colsanitas.com (L.R.-S.); paularuedag2000@gmail.com (P.A.R.); geralap10@hotmail.com (G.A.); 4Genética Molecular (GenMol), Facultad de Ciencias Exactas y Naturales, Universidad de Antioquia, Medellín 050012, Colombia; johnny.soto@udea.edu.co (A.S.-O.); andres.villegas@udea.edu.co (A.V.-L.); 5Grupo de Neurociencias de Antioquia (GNA), Facultad de Medicina, Universidad de Antioquia, Medellín 050012, Colombia; 6Laboratorio Clínico y de Patología, Clínica Colsanitas, Bogotá 110221, Colombia; magodoy@colsanitas.com; 7Grupo de Investigación Estudios en Psicología, Departamento de Psicología, Escuela de Humanidades, Universidad EAFIT, Medellín 050022, Colombia; jorgemauricio.cuartas@gmail.com; 8Universidad del Norte, Barranquilla 080001, Colombia; jvelezv@uninorte.edu.co (J.I.V.); oorjuela@uninorte.edu.co (O.M.V.); 9Grupo de Investigación en Psiquiatría (GIPSI), Departamento de Psiquiatría, Instituto de Investigaciones Médicas, Facultad de Medicina, Universidad de Antioquia, Medellín 050012, Colombia

**Keywords:** Purine-rich element binding protein A gene, PURA, transcriptional activator protein Pur-alpha, cognitive developmental delay, mental retardation, Mutation c.697_699del p.Phe233del

## Abstract

A whole-exome capture and next-generation sequencing was applied to an 11 y/o patient with a clinical history of congenital hypotonia, generalized motor and cognitive neurodevelopmental delay, and severe cognitive deficit, and without any identifiable Syndromic pattern, and to her parents, we disclosed a de novo heterozygous pathogenic mutation, c.697_699del p.Phe233del (rs786204835)(ACMG classification PS2, PM1, PM2, PP5), harbored in the *PURA* gene (MIM*600473) (5q31.3), associated with Autosomal Dominant Mental Retardation 31 (MIM # 616158). We used the significant improvement in the accuracy of protein structure prediction recently implemented in AlphaFold that incorporates novel neural network architectures and training procedures based on the evolutionary, physical, and geometric constraints of protein structures. The wild-type (WT) sequence and the mutated sequence, missing the Phe233, were reconstructed. The predicted local Distance Difference Test (lDDT) for the PURAwt and the PURA–Phe233del showed that the occurrence of the Phe233del affects between 220–320 amino acids. The distortion in the PURA structural conformation in the ~5 Å surrounding area after the p.Phe233del produces a conspicuous disruption of the repeat III, where the DNA and RNA helix unwinding capability occurs. PURA Protein–DNA docking corroborated these results in an in silico analysis that showed a loss of the contact of the PURA–Phe233del III repeat domain model with the DNA. Together, (*i*) the energetic and stereochemical, (*ii*) the hydropathic indexes and polarity surfaces, and (*iii*) the hybrid Quantum Mechanics–Molecular Mechanics (QM–MM) analyses of the PURA molecular models demarcate, at the atomic resolution, the specific surrounding region affected by these mutations and pave the way for future cell-based functional analysis. To the best of our knowledge, this is the first report of a de novo mutation underpinning a PURA syndrome in a Latin American patient and highlights the importance of predicting the molecular effects in protein structure using artificial intelligence algorithms and molecular and atomic resolution stereochemical analyses.

## 1. Introduction

The *Purine-rich element-binding protein A* gene (*PURA*; MIM: 600473) is a single exon gene harbored on the chromosome 5q31.2 that encodes the transcriptional activator protein Pur-alpha (PURA; Uniprot: Q00577), a multifunctional protein, and is a member of the Pur family of nucleic acid-binding proteins which consist of a glycine-rich flexible amino terminus, a central core region, and a potential carboxy-terminal protein binding region [1,2,3]. The *PURA* gene is expressed almost ubiquitously, including in the brain, muscle, heart, and blood [1,2,3].

All human Pur proteins have three highly conserved sequence-specific repeats (Pur repeats I–III) of 64–80 amino acids that are the hallmark of the Pur proteins [1]. Pur-alpha has the helix-unwinding capability and has been shown to bind specific sequences of ssDNA, dsRNA, and ssRNA, with a preference for GGN-repeats (i.e., the GGCGGA sequence derived from the myelin essential protein (MBP) proximal regulatory region) [4], to regulate a variety of cellular processes including DNA replication, gene transcription, RNA transport, and mRNA translation [5,6].

Pur-alpha plays a major role in developing the central nervous system acting in processes such as proliferation, dendrite maturation, and the transport of mRNA to translation sites in hippocampal neurons [4,7]. Knockout mice models of *PURA* show failure to thrive and develop neurologic features, including ataxic gait, hind limb weakness, and abnormal movements such as tremors [4,7]. 

In humans, *PURA* heterozygous mutations underlay a complex dominant phenotype (PURA syndrome) characterized by moderate to severe hypotonia; neurodevelopmental, motor, and language delay; feeding difficulties; apneas; epileptic seizures; abnormal nonepileptic movements; visual problems; and, less commonly, congenital heart defects; urogenital malformations; skeletal abnormalities; and endocrine disorders [1,2,3,8]. Since the description of severe mental retardation in four patients with de novo mutations in the *PURA* gene, up to date, ~60 different mutations have been characterized in 75 individuals with PURA syndrome [1,2,3,8,9,10,11,12,13,14,15,16].

Prediction of the functional impact of *PURA* mutations is difficult, considering the pleiotropic nature of the protein. Furthermore, PURA is expressed in almost every cell, making it more challenging to create a mutational profile [5,6]. Therefore, the characterization of nonsynonymous mutations in PURA requires further study to address the role of PURA mutations in the etiology of the disease and potential pathways affected to induce its pathogenesis.

In this article, we present the identification of a de novo heterozygous pathogenic mutation, c.697_699del p.Phe233del (rs786204835), harbored in the *PURA* gene in an 11 y/o girl with cognitive developmental delay. As this mutation has been previously described, we explore different in silico protein structure prediction tools, such as AlphaFold [17], to study the impact of this deletion. We showed that after the p.Phe233del mutation, a distortion of the ~5 Å surrounding area occurs, generating a disruption of the repeat III and a structural alteration of the β-sheet holding Phe233. We validated the interactions between PURA and the human *c-MYC* and *MBP* human upstream promotor regions. Our analysis for the predicted structures of both the AlphaFold PURA wild-type (PURAwt) and the PURA–Phe233del successfully showed DNA–protein biding impairment.

These results will pave the way for future prediction/validation of DNA and RNA binding proteins affected in human disorders through AlphaFold-predicted structures modeling and refined stereochemistry experiments at atomic resolution [18,19]. This article represents the first report of a de novo mutation underpinning a PURA syndrome in a Latin American patient and highlights the importance of predicting the molecular effects on protein structure with artificial intelligence and advanced molecular chemistry algorithms.

## 2. Patients and Methods

The propositus is an 11 y/o female, the unique daughter of nonconsanguineous parents, with a history of congenital hypotonia, failure to thrive, severe cognitive and motor deficit, and a current language capability consisting of sparse disyllables. At the physical examination in good general condition: Heart Rate: 74 beats/min. Respiratory Rate: 20 breaths/min. Systolic Blood Pressure: 95 mmHg. Diastolic Blood Pressure: 50 mmHg. Mean Blood Pressure: 65 mmHg. Temperature: 36 °C. Sense Organs: Oral mucosa moist and pink, oropharynx not congestive, without plaques or exudates, right and left eye without alterations, right and left ear without alterations. Neck: No masses. Cardiovascular: Rhythmic heart sounds, no presence of murmurs. Pulmonary: Respiratory sounds present in both pulmonary fields, without aggregates. Abdomen and pelvis: Soft, depressible, not painful, no signs of peritoneal irritation, bowel sounds present. Upper extremities: Peripheral pulses present, rhythmic, and regular. Lower extremities: Peripheral pulses present, rhythmic and regular, without edema. Osseo–muscular–articular: No alterations. Neurologic: Alert, oriented, without apparent deficits. Skin and Fanners: Normal. Paraclinical Tests: Brain MRI was reported as normal, FISH to discard Prader–Willi syndrome reported as normal.

### 2.1. Whole Exome Capture 

Three methods were applied to DNA quantification and qualification: (1) DNA purity was checked using the Nanodrop (OD260/280 ratio); (2) DNA degradation and contamination were monitored on 1% agarose gels; (3) DNA concentration was measured using Qubit. DNA samples with OD260/280 ratio between 1.8~2.0 and concentration above 1.0 ug were used to prepare sequencing libraries. 

### 2.2. Next Generation Sequencing

Next-Generation Sequencing (NGS) of the whole exome (the entire coding region of the genome and intron–exon junction regions, corresponding to approximately 23,000 nuclear genes) was performed on a DNB-SEQ400 next-generation mass sequencer using an MGI-V5 exome library. All sequenced data were quality assessed (base quality distribution, nucleotide distribution, and presence of adapters, chimeras, and other contaminants) to identify/remove low-quality data/samples from further analysis.

### 2.3. Bioinformatic Analysis 

All high-quality data were then mapped to the human genome assembly using the bwa–mem algorithm. Aligned files were processed using Genome Analysis Tool Kit (GATK) for base quality recalibration, indel realignments, and duplicate removal. This was followed by SNP and INDEL discovery and genotyping (plus phasing where applicable) according to GATK best practices recommendations [20].

All variant calls were subject to variant quality score recalibration and filtering to remove low-quality variants. The remaining high-quality variants were annotated for predicted functional consequences using the Voting Report Index, including SIFT, PolyPhen2 HVAR, Mutation Taster, Mutation Assessor, FATHMM, and FATHMM MKL Coding. For example, for a conservative filter, items with 0, 1, or maybe 2 tolerated predictions were kept. A more conservative filter would keep based on 3, 4, or 5 damaging predictions. Many variants do not have five algorithms with nonmissing values [21,22,23,24].

Subsequently, variants identified in genes with the known clinical association and related to monogenic and mitochondrial diseases were analyzed in contrast to our patient’s phenotype and clinical manifestations. Only genes with an average coverage of more than 98% and a minimum depth of 20× were analyzed for accomplishing quality standards. As a note of caution, it is essential to mention that this study may not cover all possible pathogenic variants in each gene, and for technical and scientific reasons, whole-exome sequencing may not completely cover the entire coding region of the human genome. The sequencing results were bioinformatically analyzed using the DreamGenics-GenomeOne software, and the aligned and filtered sequences (fulfilling specific quality criteria) were compared against the GRCh37/hg19 reference genome for annotation and variant calling.

Bioinformatics analyses aimed to identify variants included in exonic regions or splicing regions (at least 20 bp), small insertions and deletions, and, in particular cases, genes related to the patient’s clinical phenotype. Parameters such as the population frequency of the variant (i.e., in ExAC, gnomAD, and 1000 Genomes databases), its allelic influence, the associated OMIM phenotype, its classification in ClinVar, and the inheritance model were used to filter the variants to be reported. This sequential, comprehensive, and sometimes targeted analysis allowed for the identification of exonic deletions and duplications (also known as Copy Number Variants, or CNVs) and variants involving large gene regions. If CNVs were identified, a secondary confirmation method was either applied or recommended, as NGS is an indirect method for their identification [21,22,23,24]. 

Variants outside of coding regions or harbored out of the intron–exon junctions of the gene cannot be identified (e.g., variants in promoter regions, enhancers, regulatory regions far from the exonic region, unique variants including dynamic mutations, complex recombination structures, structural variants of genes (e.g., inversion-type rearrangements), and epigenetic effects). Identified variants have been evaluated against HGMD, ClinVar, LOVD, dbSNP, and gnomAD databases. For missense and splicing site variants of uncertain significance, in silico prediction tools were used, and their analysis is reported in the interpretation of the results. Finally, the association of the identified variants with the syndromes described in OMIM and the clinical association with the phenotype described in the patient are evaluated [21,22,23,24].

We followed the recommendations of the American College of Medical Genetics and Genomics (ACMG) and reported pathogenic or probably pathogenic variants in 59 clinically significant genes [25,26,27,28]. According to ACMG and the current knowledge, mutations identified as benign are not reported; these variants, in general, have an allele frequency greater than or equal to 1% and result in a synonymous amino acid change or occur in 5′ or 3′ untranslated regions. This information can be made available upon request. A negative result does not rule out the possibility that the tested individual has a rare, untested mutation in an undetectable region. In these cases, reanalyzing the data in the future could generate new results.

### 2.4. Protein Reconstruction

We used the significant improvement in the accuracy of protein structure prediction recently implemented in AlphaFold, which incorporates novel neural network architectures and training procedures based on the evolutionary, physical, and geometric constraints of protein structures. Furthermore, 3D protein structure reconstruction with AlphaFold is vastly more accurate than those obtained by competing methods, i.e., median backbone accuracy, highly accurate side chains reconstruction, accurate domains, domain-packing prediction, and precise, per-residue estimates of its reliability [17,19,29]. The wild-type (WT) sequence (Uniprot: Q00577) and the mutated one, missing the phe233, were reconstructed. 

#### 2.4.1. PURA Protein–DNA Docking in Silico Analysis

We applied HDOCK (http://hdock.phys.hust.edu.cn/ (accessed on 22 June 2022)), a highly integrated suite for several in silico features such as macromolecular docking. HDOCK supports protein–RNA/DNA docking with an intrinsic scoring function. This tool, based on a hybrid algorithm of template-based modeling and ab initio free docking, delivers both template- and docking-based binding models of two molecules and allows interactive visualization [30]. A PURA sequence-specific, single-stranded DNA-binding was reported in cell culture and brain extracts [31,32,33,34]. We used these reported purine-rich sequences found within a zone to initiate DNA replication upstream of the human *c-MYC* and MBP genes [35].

We used the *c-Myc* upstream region Genomic chr8: 5′ tttctcttttggaggtggtggagggagagaaaagtttacttaaaatgcct 3′ and MBP upstream region Genomic chr18, specifically, 5′ cagggagccgcccccacttgatccgcctcttttcccgagatgccccggggagggaggacaacaccttcaaagacaggccctctgagtccgacgagctcca 3′. 

#### 2.4.2. Energetic and Stereochemical Characterization of the PURA Molecular Models

Models obtained by molecular modeling were energetically and stereochemically characterized. For the energetic characterization, we used SAVES v6.0 coupled to the ERRAT parameter. Additionally, we used the SWISS_MODEL server to characterize the QMEAN6 parameter to define a normalized value (linear combination of six terms in favor of its stability). We also estimated the *Z*-score parameter to indicate whether the model is in trend or not in comparison with experimentally solved structures (either with X-ray diffraction [XRD], cryogenic imaging microscopy [CryoEM], or solid nuclear magnetic resonance [NMR] techniques) [36,37]. The stereochemical characterization was performed with the PROCHECK software, which allows one to measure the stereochemical viability of the model by considering the dihedral angles Phi (φ) and Psi (ψ) of the planes generated by the peptide bonds of the amide functional groups that link the amino acids in the proteins. It performs all this with the α-carbons that constitute the mobility in the protein system [38].

#### 2.4.3. Calculation of Hydropathic Indexes and Polarity Surfaces of PURAwt and Mutated-Phe233del

Because a deletion with potential effects was found on the conformation of the secondary structure, polarity evaluation was performed through the hydropathicity index, and the Molecular Lipophilicity Potential (MLP) maps for proteins. The polarity changes in the surroundings of the mutation were calculated with the Protscale software of the Swiss ExPASY suite, which compares the primary sequence of the PURAwt and the mutated-Phe233del variant using the Kyte and Dolittle coefficients. Depending on the score, and its sign, a hydrophilic environment (negative) or a hydrophobic environment (positive) can be favored, with zero being the threshold that separates them [39]. The molecular surface lipophilicity potential was obtained with the Chimera software, and was categorized in a range between (−20, 20) defined with a color code; negative hydrophilic values are represented in green, while the positive hydrophobic are represented in gold [40].

#### 2.4.4. 3D Alignment Structural Viewers

This 3D alignment is intended to obtain the coincidence structurally in a significant part of the spatial distribution of both models. The spatial co-ordinates and the selection of the molecular regions were made with the Deepview/Swiss-PdbViewer software version 3.7 [41]. The Chimera UCSF v 1.1.1 software was used for visualization, and the structural alignments were made with the Needleman–Wunsch global algorithm and the BLOSUM62 matrix. The quantitative alignment to find the RMSD value in the structural comparison was made using PyMol [40,42,43].

#### 2.4.5. Simulation by the Hybrid Quantum Mechanics–Molecular Mechanics QM–MM Method

To better quantify the Phe233 mutation effects at the atomic functional resolution, we applied a combination of hybrid quantum chemistry theory (focusing on the deletion vicinity region) and molecular mechanics force field theory (applied to other PURA regions far from the vicinity of the Phe233 deletion). Both theories amalgamated in the QM/MM semiempirical method (preferred for macromolecular systems) [44,45,46,47,48]. We used the model’s parameterization with the *Z* matrix of connectivity and experimental data derived from empirical crystallographic studies, focusing on those targeting the Phe233del position. Thus far, the quantum analysis (QM) focuses on the antiparallel β-sheets of the secondary structure, which contemplates the amino acids Asn228-Thr252 for the wild type and Asn228-Tyr252 for the mutated structure. This region was delimited from the saturation with hydrogen atoms using the Austin model1 (AM1) (base of the software spartan20’ of Wavefunction acquired license, https://www.wavefun.com/corporate/more_spartan.html (accessed on 22 June 2022)) to measure, from the electronic correlation, changes of bond distances and angles, as well as energetic changes [48,49,50]. The QM region also undergoes a molecular mechanics (MM)_QM_ approach with the MMFF_aq_ forcefield to better understand the second solvation sphere, given that the surrounding targeted region is a polyatomic system. The rest of the structure undergoes an approach with classical physics from molecular mechanics (MM) with the MMFF_aq_ base, which favors systems that can be measured in terms of the properties from the electronic correlation at the atomic level, i.e., energy changes, bond distances, and changes in the angles [49,51]. In the simulation, a geometric optimization was performed, and the lowest energy conformer was obtained, which serves as a model for calculating the surface properties of the electronic structure (electrostatic potential maps in ranges from −200 kJ/mol to 200 kJ/mol). This allows us to understand the electronic distribution and the possibility of interacting or anchoring with noncovalent interactions in the multiscale model [49,50,51]. 

## 3. Results 

### 3.1. WES Bioinformatics Identified a De Novo Mutation in PURA

We identified a de novo (not present in the parents) heterozygous mutation in the proband, c.697_699del (p.Phe233del), NM_005859.5, PURA (MIM*600473) (Het, Autosomal dominant mental retardation 31) (MIM #614563), located at Chr5:139,494,453-139,494,455 (rs786204835) (additional data regarding next-generation sequencing quality parameters are presented in Appendix A). This deletion of a complete codon located from position 697 to 699 of the cDNA causes the deletion of a phenylalanine at residue 233 of the protein (inframe), which, according to predictions, affects a functional domain of the protein (Figure 1a).

### 3.2. The 3D Reconstruction of the PURAwt Structure Showed Significant Defects Caused by the Phe233 Mutation

The prediction model generated by the AlphaFold resulted in a very high per-residue estimate of its confidence (pLDDT) score > 90 within the Phe233 area (β-sheet), as presented in the β-sheet composition with three Phenylalanine’s expansion (Figure 1b). The PyMol software showed the composition of the β-sheet containing the Phe233. In agreement with protein–DNA binding interactions, protein structures, such as the protein region containing the deletion, may induce a deformation of the DNA due to binding in specific regions of the DNA [52]. For instance, it is binding to the minor groove that, in addition to electrostatic interactions and other noncovalent interactions, may also play an essential role in generating such deformation. This will allow the transcription initiation to generate mRNA, i.e., AlphaFold analysis of the region harboring the Phe233 change showed its hydrogens in a 3D conformation (Figure 1c) and the Phe233 polar contacts measurements were less than 4 Å (Figure 1d). Appendix A summarizes the evaluation parameters and scores obtained by different software on the available models.

Additionally, a global alignment with the Needleman–Wunsch algorithm and a standard deviation value (RMSD = 0.595) was performed by zooming in on the region near position 233 of the wild-type R245-D234 protein region (translated from the PURA gene) to evaluate the effect of the deletion and potential changes in the angle of the β-sheets of PURAwt once the PURA–Phe233del occurred (Appendix A). The acid-base interactions folding the β-sheets with a hydrogen bond molecular interaction at a distance of 1.911 Å in the wild-type protein (Appendix A) favors interactions between the aspartic acid with arginine by a nonbonding acid-base interaction between R245–D233, decreasing the distance to 1.768 Å, once the deletion is present (Appendix A).

#### 3.2.1. PURA Protein–DNA Docking in Silico Analysis Revealed Significant Effects of the Phe233 Mutation Affecting Interactions with the DNA

The modeling of the PURAwt protein by AlphaFold and posterior visualization of the three-principal purine-rich element-binding (PUR) interacts with its three PURA RNA/DNA binding domains (Figure 2a). It shows a spanner head containing the c-Myc DNA sequence of three different HDOCK-generated docking models from the top 10 (see DNA in dark blue, light blue, and white). Residues responsible for DNA contact are marked in white (see white arrows) (Figure 2b,c). See, specifically, the modification of the residues marked in white, losing contact with the DNA, in the case of the PURA–Phe233del model when compared to the PURAwt.

Predicted PURAwt and PURA–Phe233del by AlphaFold were docked to previously reported DNA-promoter regions of the *c-MYC* and *MBP* human genes [31,32,35]. In silico HDOCK results validated previously experimentally reported interactions between PURA and the *c-MYC* and *MBP* DNA promoter region sequences. The top ten in silico models aligned perfectly within the PURAwt model (Figure 3a), whereas docking between PURA–Phe233del and *c-MYC* and *BMP* sequences showed a disruption in the top 10 HDOCK in silico models (Figure 3b). 

#### 3.2.2. Energetic and Stereochemical Characterization of PURA Molecular Models Indicate Protein Damage

The structures at the energetic level have QMEAN6 values within the normalized ranges of (0–1), presenting larger energetic values for PURAwt of 0.61 compared to the structure with the deletion of 0.59. Likewise, they have *Z*-score values close to the scattering values of the structure plot, contemplating experimentally solved structures with values of −2.41 (PURAwt) and −3.60 (PURA Phe233del). Additional data is shown in Appendix A.

Considering the statistical values and the structure of amino acids for the local groupings of nonbonding interactions, the Errat software estimated high scores, close to 93.89% for PURAwt and 89.66% for PURA Phe233del. At the stereochemical level, measurements of the peptide bonds alpha–carbons–dihedral angles (Cα) using the Ramachandran plots showed, for the PURAwt, 219 amino acids in the favored region (84.6%), 23 in the allowed region (8.9%), 4 in the generously allowed region (1.5%), and 23 in the disallowed region (5.0%). In contrast, for the PURA Phe233del, there are 216 amino acids in the favored region (83.7%), 30 in the allowed region (11.6%), 3 in the generously allowed region (1.2%), and 9 in the disallowed region (3.5%) (Appendix A).

The model was chosen because, for the AlphaFold prediction model, either for the PURAwt or PURA Phe233del, the best-obtained values maximize energetic level, protein size, and stereochemistry with a lower percentage in the disallowed regions. In the case of PURAwt, the number of residues in the disallowed region was 13, corresponding to Proline and Glycine residues harboured in the loops of the secondary structure. On the other hand, for PURA–Phe233del, the number of residues in the disallowed region was 9, and were also constituted by Proline and Glycine residues. 

#### 3.2.3. Calculation of Hydropathic Indexes and Polarity Surfaces of PURAwt and PURA–Phe233del Suggests Changes in Protein Structure

The lipophilicity potential structures showed that Phe233 is harbored in a golden region (hydrophobic), indicating that it plays a significant role in that three-dimensional arrangement (Figure 4a). Once the Phe233del occurs, the lipophilicity potential (polarity of the protein environment) changes, so the whole environment near the deletion is made green, and the polarity increases, which induces a change in the structure (Figure 4b). It can also be observed that the polarity increased in the vicinity of the deletion since more negative values of the environment are obtained for the Kyte and Dolittle coefficients. Indeed, the polarity is −0.056 at position PURAwt, which changes to −0.756 (Asp233) (Figure 4c).

#### 3.2.4. The Hybrid Quantum Mechanics–Molecular Mechanics (QM–MM) Indicates That the Binding Distance of the PURAwt Decreases to 6.103 Å, with the Phe233 mutation which Affects the Accessible Area of the Structure

For the PURAwt, the estimated bond angle between amino acids F232–F233 was 37.57°, while for F233–D234, a bond angle of 113.84° was obtained, which separates the reference points on the CH2 carbons constituting the side chain by 7.215 Å (reference distance of the amino acids in the β-sheets) (Figure 5a). Once the deletion occurs, the bond angles between F231–F232 are 35.51°, and the bond angle between F232–D233 changes to 112.47°, indicating that for the reference structure, the binding distance decreases to 6.103 Å, which affects the accessible area of the structure (Figure 5b). The electrostatic potential maps were calculated for the β-sheets involved in the region undergoing variation, and area and volume values for the electronic region were obtained; the area and volume values for the electronic region analyzed using quantum chemistry were 3032.37 Å^2^ and 2930.94 Å^3^, respectively (Figure 5c). Upon the occurrence of the Phe233 deletion, the area value in the electronic region changes to 2852.30 Å^2^ and the volume to 2773.77 Å^3^, which is in agreement with the loss of enough electronic topology contributed by the Phe233 amino acid and its aromaticity (Figure 5d).

## 4. Discussion

Variants in the *PURA* gene have been linked to autosomal dominant mental retardation 31, which has an autosomal dominant inheritance pattern and correlates with the patient’s phenotype. This de novo variant fits the PS2 criteria of the ACMG guidelines and is an in-frame deletion of a highly conserved residue of a functional domain of the protein (PM1) that has no allele frequency (PM2).

The PURA protein was first described to bind to ssDNA in cell culture and brain extracts [31,35]. PURA was shown to play a role in cellular transcriptional activity [31,35]. The *PURA* gene has been reported as part of the 3% of human genes that lack introns [8]. It seems that this gene cannot generate alternative splicing. Evolution has tagged this genomic region as inalterable, which may prove its unique and essential role in interacting with DNA and RNA molecules. It has been reported that any mutation in the PURA protein causes the full spectrum of the human PURA syndrome [53].

PURAwt and PURA Phe233del prediction models derived with AlphaFold showed a close molecular interaction between the Phenylalanine’s molecules. Hence, changes in this structural conformation may alter PURA protein’s binding and transcription initiation capacities. This may suggest that salt-bridge interactions between the phosphate groups and Lys or Arg residues, along with the intercalation of Phe residues between two base-pair stacks, stabilize and open-up DNA conformation, which can be altered in the PURA Phe233del mutation. Furthermore, the design of superhelical nanostructures by minimal molecular elements has demonstrated the importance of Phenylalanine as a molecular block, establishing the structure of the phenylalanine zipper, which consists of aromatic side chains from ten phenylalanine residues that are stacked within a hydrophobic core [19]. This zipper has been demonstrated to mediate the dimerization of various proteins, including APS, SH2-B, and Lnk [20].

The analysis of the AlphaFold prediction by the Local Distance Difference Test (pLDDT), for the PURAwt and the PURA–Phe233del predicted structure deletion per amino acid, using five different models via AlpahFold2 models, showed that the occurrence of the Phe233del affects between 220–320 amino acids (red arrow) (Figure 6a,b). Additionally, the 3D reconstruction of the transcriptional activator protein Pur-alpha (PURA; Uniprot: Q00577) from the wild sequence using Alpha fold, and the 3D reconstruction of the PURA–Phe233del (Figure 6c,d), with a focus on the surrounding area of the Phe233 amino acid (in magenta), disclose the distortion in the PURA structural conformation after the PURA–Phe233del occurs, with a conspicuous disruption of the repeat III.

The protein PURA has been shown to bind DNA and RNA, implicated in these molecules’ transcriptional and intracellular localization. PURA protein molecular variations have been linked to several neurodevelopmental disorders [8,44,45]. In this manuscript, we also showed that the HDOCK modeling of PURAwt aligns perfectly with the *c-Myc* DNA promotor region, whereas alteration in PURA-Phel233del causes an impairment in the DNA binding capacity of PURA (Figure 7a,b).

These results provide new evidence to validate previous reports, which is significant considering that most studies of PURA molecular interaction were performed before the X-ray structure of PURA was available. Moreover, reported interactions were disclosed with methods that did not distinguish between direct or indirect interactions [54]. Thus far, in silico protein prediction and molecular interaction tools such as docking are essential to further investigate whether those interactions represent direct binding events and identify their physiological relevance.

The QM–MM simulation allowed us to analyze the structural changes of PURAwt and PURAdel233Phe by dissecting, at atomic resolution, the electronic structure restricted to the targeted region where the mutation occurred. We used a semiempirical theoretical approach to the macromolecular level with an Austin Model1, which allows us to measure properties such as (*i*) angle change and (*ii*) surface potential; the rest of the PURA protein structural changes, due to computational costs, are described with classical physics from the MMFF basis. The changes in area/volume and angular twists imply steric changes that affect the functionality of the protein and its interaction; therefore, twists or reductions in area and volume space mean an interaction impairment with molecules such as nucleic acids, which is diminished with the deletion of an electronically bulky group such as the Phe233. 

The energetic and stereochemical molecular modeling of the PURAwt and the mutated PURA–Phe233del showed good scores. The QMEAN6 is a method conceived as a linear combination of stability terms involving energy of the torsion angles, coupling energy of atom pairs, solvation energy, beta carbon energy, secondary structure agreement, and solvent accessibility agreement. These parameters generated a high global value for the two structures in terms of stability. Likewise, the *Z*-score parameter is a term that allows a comparison of the predicted hypothetical models to those experimentally obtained by X-ray diffraction techniques with a monochromator, NMR nuclear magnetic resonance, and CryoEM cryogenic imaging microscopy. We can observe that both structures leave the model located in the trend of the graph, as shown in Appendix A. The Errat software has a statistical estimation algorithm that allows for the grouping of nine amino acids and performs a characterization, following along the whole primary sequence, to generate the summation of a global value of the proteins, producing a very high score for the two hypothetical models obtained.

In stereochemical terms, the measurement of the spatial behavior of the models was made from the dihedral angles considering the alpha carbons (Cα), belonging to the side chain, measuring the residues belonging to specific regions within the Ramachandran graph, dependent on the twists and angles of the secondary structure, they are adopting. These values depend on the movement of the angles Phi (φ) and Psi (ψ), whose rotations are measured with forbidden rotations by the amino acids glycine and proline located in the N-terminal and C-terminal loops. Likewise, the values obtained show a total percentage for PURAwt of 95% in the favored regions and 96.5% for PURA–Phe233del since it loses an amino acid in an antiparallel beta-sheet folding. This occurs because it has hydrogen bonding interactions every 180° for every 1–3 amino acid that belongs to that secondary structure and becomes a flexible loop, releasing forbidden conformations.

After analyzing the polarity changes and their effect at the structural level (Kyte and Dolittle coefficients) and using the Needleman–Wunsch algorithm to visualize the effect at the 3D level, we concluded that a twisting effect, mediated by the Phe233del deletion, produces a practical approach of the antiparallel β-sheets by noncovalent interactions, i.e., hydrogen bonding that decreases the interaction distance with the variant, causing a structural torsion. Consequently, these chemical effects favor the formation of adducts (the interaction distance between the amino acids D233 and R245 is decreased), causing a change in the torsion angle in the F232–D233 peptide bond. This topological effect alters the electronic surface of the antiparallel β-sheet (since an amino acid with a high electron density is lost) that provides more access to ligands, allowing the recognition of the nucleotide chains associated with the III region of the DNA recognition domain. Our conclusion is corroborated by the area and volume decrease of 175.07 Å^2^ and 157.17 Å^3^, respectively. This change is quantified with the accessible area that a ligand can have, and this exposed area goes from 1071.62 Å^2^ in the case of the PURAwt to 990.97 Å^2^ in the case of the PURA–Phe233del mutated protein, which affects the protein functionality. 

## 5. Conclusions

This manuscript presents a first report of a de novo mutation underpinning a PURA syndrome in a Latin American patient. We predicted the molecular effects in the protein structure using artificial intelligence algorithms with AlphaFold, a novel neural network. The p.Phe233del mutation produces a distortion in the PURA structural conformation in the ~5 Å surrounding area of the repeat III, where PURA interacts with DNA and RNA helix unwinding. Further to Protein–DNA docking analysis, we used energetic and stereochemical and the hybrid Quantum Mechanics–Molecular Mechanics (QM–MM) analyses to achieve an atomic resolution of the specific surrounding region affected by the mutation. These analyses are essential in understanding the molecular effects in protein structure and the efficiency of artificial intelligence algorithms and molecular and atomic resolution stereochemical analyses. 

## Figures and Tables

**Figure 1 brainsci-12-00871-f001:**
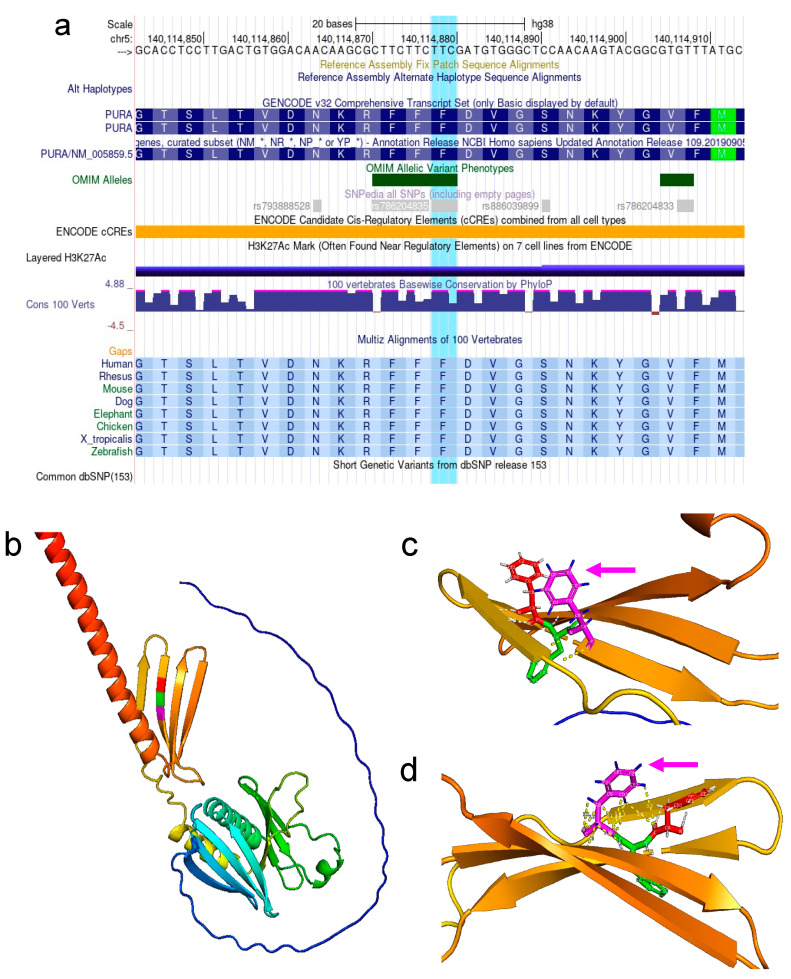
(**a**) De novo heterozygous mutation found in the proband, c.697_699del (p.Phe233del), NM_005859.5, PURA (MIM*600473), located at Chr5:139,494,453-139,494,455 (rs786204835). This deletion causes the loss of Phenylalanine at residue 233 of the protein (in-frame), codon located from position 697 to 699 of the cDNA. (**b**) β-sheet composition with three Phenylalanine expansions: Phe231 (red), Phe232 (green), and Phe 233 (magenta). (**c**) Phe233 shows its hydrogens in a #D conformation (magenta arrow). (**d**) Phe233 polar contact measurements are less than 4 Å (magenta arrow).

**Figure 2 brainsci-12-00871-f002:**
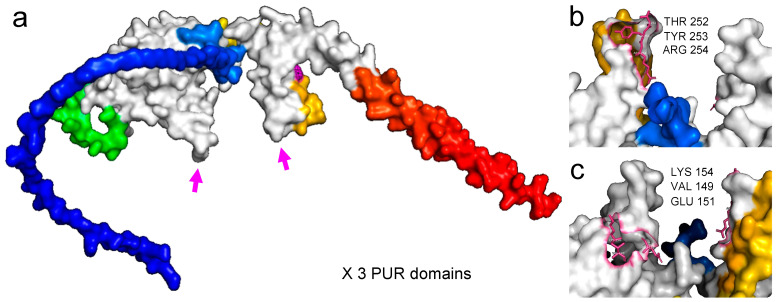
(**a**) PURAwt AlphaFold modeling showing the three principal purine-rich (PUR) elements (in white). Magenta arrows show the DNA docking site regions forming a spanner head that will interact with the DNA. (**b**) Zoom of the protein–DNA docking site with three principal residues in contact with the DNA molecule (*c-MYC* promoter region), THR 252, TYR 253, and ARG 254. (**c**) Zoom and 180-degree rotation to visualize the opposite protein–DNA docking site with its three principal residues in contact with the DNA molecule (*c-MYC* promoter region), VAL 149, GLU 151, and LYS 154.

**Figure 3 brainsci-12-00871-f003:**
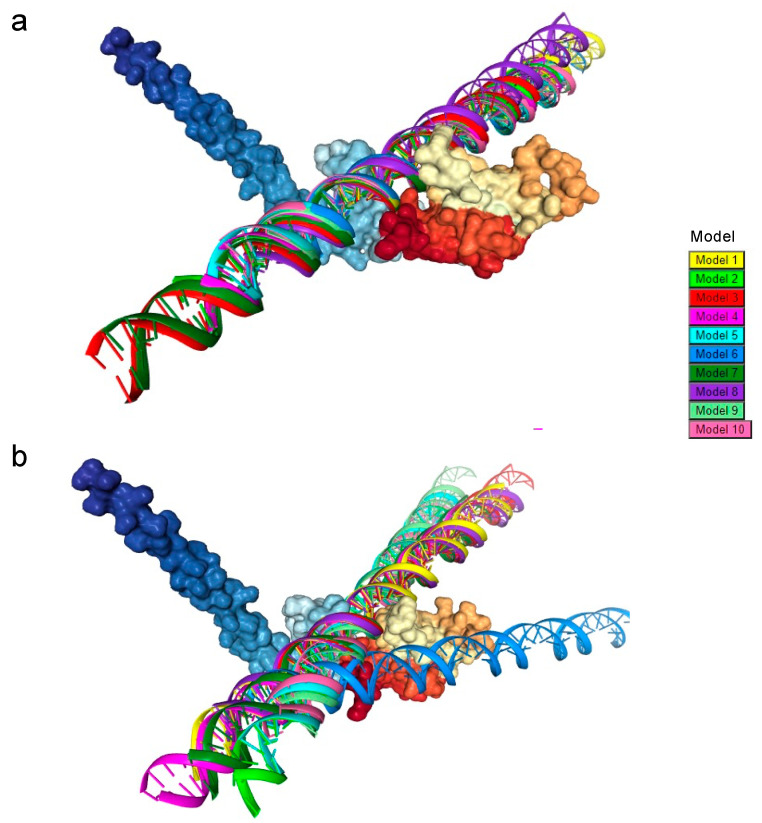
The PURA–DNA comparative analyses show that DNA docking damages PURA Protein–DNA docking. (**a**) In silico analysis by HDOCK shows 10 top models between PURA-WT AlphaFold and previously reported DNA promoter regions of the *c-Myc*. (**b**) HDOCK docking analysis between AlpahFold predicted PURA–Phe233del protein and the *c-Myc* DNA promoter regions.

**Figure 4 brainsci-12-00871-f004:**
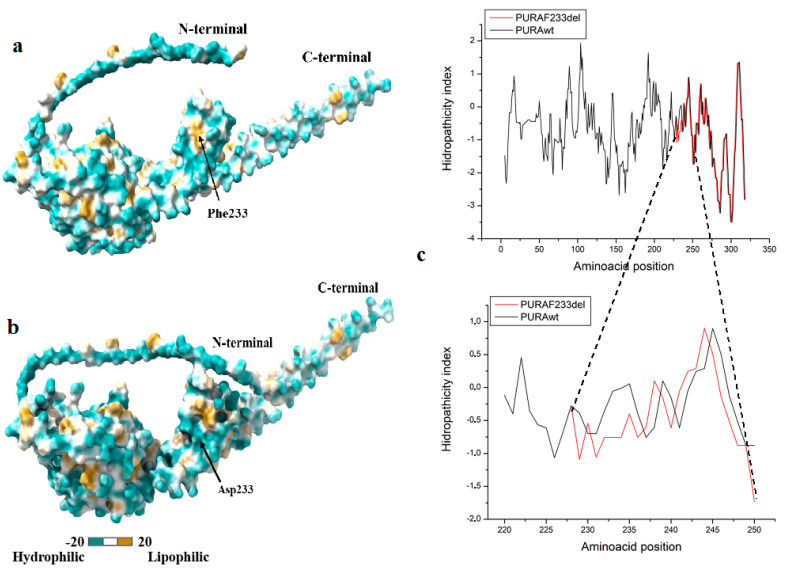
Hydrophobicity analysis (Molecular Lipophilicity Potential, MLP maps) of the (**a**) PURAwt, and (**b**) PURAPhe233del system. The analysis shows that in the wild type variant, the Phe 233 is harbored in a golden region (hydrophobic), implicating that it plays a significant role in the 3D arrangement, while the hydrophobicity changes once the deletion occurs. This is also appreciated in (**c**), which depicts the hydrophobicity index plot zooming in on the 233-amino-acid region (see text).

**Figure 5 brainsci-12-00871-f005:**
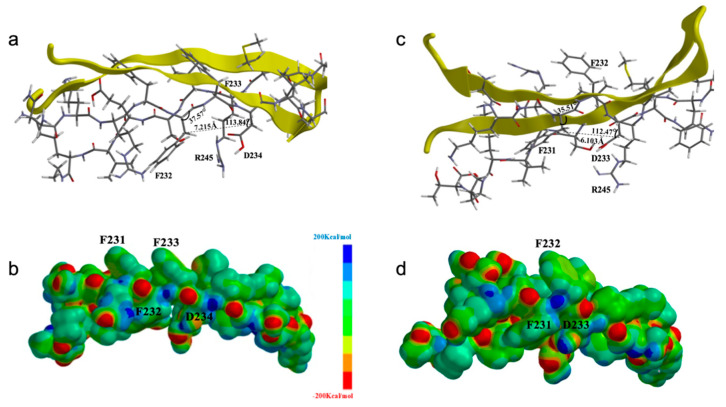
Functional comparison of the connective and electronic structures for the QM/MM simulation of PURAwt and PURA–Phe233del. (**a**) For PURAwt, the angle between amino acids F232-F233 was 37.57°, while for F233-D234, the bond angle distancing the CH2 carbons constituting the side chain by 7.215 Å was 113.84° (reference distance of the amino acids in the β-sheets). (**b**) For PURA–Phe233del, the bond angle between F231-F232 is 35.51°, and the bond angle between F232-D233 changes to 112.47°, indicating that the binding distance for the reference structure decreases to 6.103 Å, which affects the accessible area of the structure. (**c**) Electrostatic potential maps of the PURAwt β-sheets involved. The area and volume values for the electronic region in the region undergoing variation were 3032.37 Å^2^ and 2930.94 Å^3^, respectively. (**d**) Electrostatic potential maps of the PURA–Phe233del β-sheets. The area and volume values for the electronic region in the region undergoing variation change to 2852.30 Å^2^ and 2773.77 Å^3^, respectively, which is in agreement with the loss of enough electronic topology contributed by the Phe233 amino acid and its aromaticity.

**Figure 6 brainsci-12-00871-f006:**
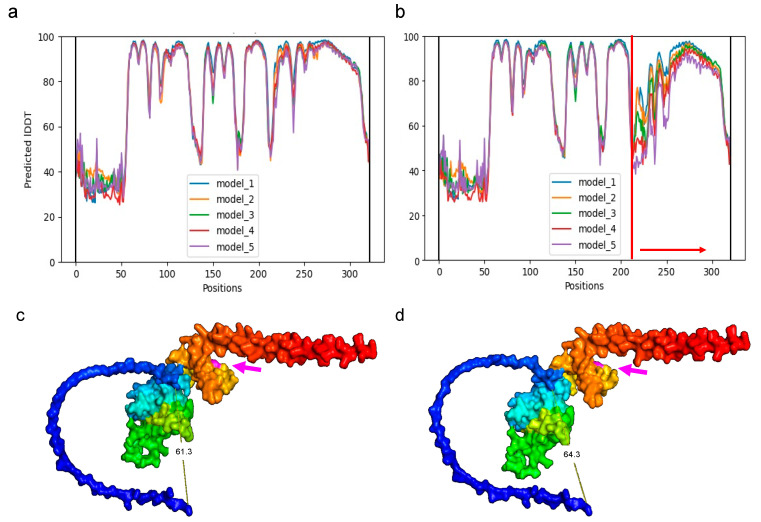
(**a**) Predicted local Distance Difference Test (lDDT) for the PURA WT and (**b**) PURA–Phe233del-predicted structure deletion per amino acid using five different models via AlpahFold2. Models show that the occurrence of the Phe233del affects between 220–320 amino acids (red arrow). (**c**) A 3D reconstruction of the transcriptional activator protein Pur-alpha (PURA; Uniprot: Q00577) from the WT sequence using AlphaFold, focusing on the surrounding area of the Phe233 amino acid (in magenta; magenta arrow). (**d**) A 3D reconstruction of the PURA–Phe233del. Note the distortion (61.3 Å vs. 64.3 Å) in PURA structural conformation after the p.Phe233del occurs.

**Figure 7 brainsci-12-00871-f007:**
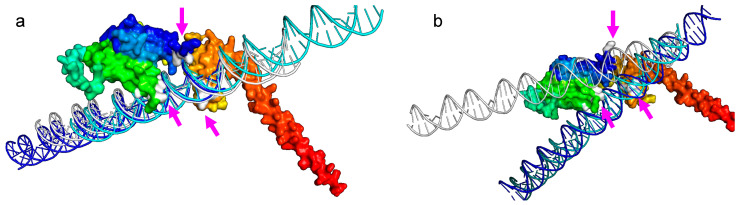
PURA–Phe233del disrupts DNA biding. (**a**) A molecular view of PURA Protein–DNA docking analysis showing three different HDOCK models between PURAwt AlphaFold and the *c-Myc* DNA promoter region. Magenta arrows indicate disruption of specific docking sites (in white; magenta arrows). (**b**) HDOCK docking analysis shows the *c-Myc* DNA binding disruption caused by the AlpahFold2 predicted PURA–Phe233del protein.

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
