# Peer review of "Structural Protein Effects Underpinning Cognitive Developmental Delay of the PURA p.Phe233del Mutation Modelled by Artificial Intelligence and the Hybrid Quantum Mechanics–Molecular Mechanics Framework"

_brainsci, 2022, doi:10.3390/brainsci12070871_

Round 1

Reviewer 1 Report

1.      For the statement in line number 95-96 “future prediction/validation of DNA and RNA binding proteins affected in human disorders through AlphaFold-predicted structures modeling and refined experiments of stereochemistry at atomic resolution.”, authors may present previously done supportive work or reference.

2.      For the methods section 2.3, 2.4, authors should add some references utilizing the mentioned tools and methods for experimentation. It would be interesting for readers.

3.      In Figure 6, Author may mention the distances after and before deletion of Phe233 and also mention the residues name and number for clear understanding to the readers.

4.      Author needs to add the conclusion section in the manuscript. This is helpful for readers.

Author Response

Comments from Reviewer #1

  1. For the statement in line number 95-96 “future prediction/validation of DNA and RNA binding proteins affected in human disorders through AlphaFold-predicted structures modeling and refined experiments of stereochemistry at atomic resolution.”, authors may present previously done supportive work or reference.

We introduced two references citing recently published research articles showing results of functional predictions of mutations while using a methodological approach similar to that used in our manuscript.(1,2)

  1. Al-Numan HH, Jan RM, Al-Saud N bint S, Rashidi OM, Alrayes NM, Alsufyani HA, et al. Exome Sequencing Identifies the Extremely Rare ITGAV and FN1 Variants in Early Onset Inflammatory Bowel Disease Patients. Frontiers in Pediatrics [Internet]. 2022;10. Available from: https://www.frontiersin.org/article/10.3389/fped.2022.895074
  2. Buel GR, Walters KJ. Can AlphaFold2 predict the impact of missense mutations on structure? Vol. 29, Nature Structural and Molecular Biology. 2022.

  1. For the methods section 2.3, 2.4, authors should add some references utilizing the mentioned tools and methods for experimentation. It would be interesting for readers.

Several references were introduced in those two sections as follows:

  1. Senior AW, Evans R, Jumper J, Kirkpatrick J, Sifre L, Green T, et al. AlphaFold. Nature. 2020;577(7792).
  2. Al-Numan HH, Jan RM, Al-Saud N bint S, Rashidi OM, Alrayes NM, Alsufyani HA, et al. Exome Sequencing Identifies the Extremely Rare ITGAV and FN1 Variants in Early Onset Inflammatory Bowel Disease Patients. Frontiers in Pediatrics [Internet]. 2022;10. Available from: https://www.frontiersin.org/article/10.3389/fped.2022.895074
  3. Buel GR, Walters KJ. Can AlphaFold2 predict the impact of missense mutations on structure? Vol. 29, Nature Structural and Molecular Biology. 2022.
  4. McKenna A, Hanna M, Banks E, Sivachenko A, Cibulskis K, Kernytsky A, et al. The genome analysis toolkit: A MapReduce framework for analyzing next-generation DNA sequencing data. Genome Research. 2010;
  5. Johar AS, Anaya JM, Andrews D, Patel HR, Field M, Goodnow C, et al. Candidate gene discovery in autoimmunity by using extreme phenotypes, next generation sequencing and whole exome capture. Autoimmunity Reviews. 2015.
  6. Johar AS, Anaya JM, Andrews D, Patel HR, Field M, Goodnow C, et al. Candidate gene discovery in autoimmunity by using extreme phenotypes, next generation sequencing and whole exome capture. Autoimmunity Reviews. 2015.
  7. da Fonseca ACP, Mastronardi C, Johar A, Arcos-Burgos M, Paz-Filho G. Genetics of non-syndromic childhood obesity and the use of high-throughput DNA sequencing technologies. Journal of Diabetes and its Complications. 2017.
  8. Landires I, Núñez-Samudio V, Fernandez J, Sarria C, Villareal V, Córdoba F, et al. Calpainopathy: Description of a novel mutation and clinical presentation with early severe contractures. Genes (Basel). 2020;11(2).
  9. Paz-Filho G, Boguszewski MCS, Mastronardi CA, Patel HR, Johar AS, Chuah A, et al. Whole exome sequencing of extreme morbid obesity patients: Translational implications for obesity and related disorders. Genes (Basel). 2014;
  10. Tayeh MK, Gaedigk A, Goetz MP, Klein TE, Lyon E, McMillin GA, et al. Clinical pharmacogenomic testing and reporting: A technical standard of the American College of Medical Genetics and Genomics (ACMG). Genet Med. 2022;24(4).
  11. Kearney HM, Thorland EC, Brown KK, Quintero-Rivera F, South ST. American College of Medical Genetics standards and guidelines for interpretation and reporting of postnatal constitutional copy number variants. Genetics in Medicine. 2011;13(7).
  12. Richards S, Aziz N, Bale S, Bick D, Das S, Gastier-Foster J, et al. Standards and guidelines for the interpretation of sequence variants: A joint consensus recommendation of the American College of Medical Genetics and Genomics and the Association for Molecular Pathology. Genetics in Medicine. 2015;17(5).
  13. Kalia SS, Adelman K, Bale SJ, Chung WK, Eng C, Evans JP, et al. Recommendations for reporting of secondary findings in clinical exome and genome sequencing, 2016 update (ACMG SF v2.0): A policy statement of the American College of Medical Genetics and Genomics. Genetics in Medicine. 2017;19(2).
  14. Jumper J, Evans R, Pritzel A, Green T, Figurnov M, Ronneberger O, et al. Highly accurate protein structure prediction with AlphaFold. Nature. 2021;596(7873).

  1. In Figure 6, author may mention the distances after and before deletion of Phe233 and also mention the residues name and number for clear understanding to the readers.

We very much appreciate the reviewer’s comment. The revised version of Figure 6 now includes the distance in Å for the distortion in 3D protein structure after the p.Phe233del occurs.

  1. Author needs to add the conclusion section in the manuscript. This is helpful for readers.

We added the next Conclusion section:

Conclusion

This manuscript presents a first report of a de novo mutation underpinning a PURA syndrome in a Latin American patient. We predicted the molecular effects in the protein structure using artificial intelligence algorithms with AlphaFold, a novel neural network. The p.Phe233del mutation produces a distortion in the PURA structural conformation in the ~5Å surrounding area of the repeat III, where PURA interacts with DNA and RNA helix unwinding. Further to Protein-DNA docking analysis, we used energetic and stereochemical and the hybrid Quantum Mechanics-Molecular Mechanics (QM-MM) analyses to achieve an atomic resolution of the specific surrounding region affected by the mutation. These analyses are essential to understanding the molecular effects in protein structure and the efficiency of artificial intelligence algorithms and molecular and atomic resolution stereochemical analyses.

Reviewer 2 Report

There are following things which needs to be improved for the easy readability and understanding of the readers. 

1) Authors might consider improving the quality of figures, and it still needs significant improvement, so that it would be clear to understand. 

2) All the results headings look like methods headings e.g. "3.2. Protein Reconstruction ", author(s) would write the main finding highlights as one sentence headings of each result paragraph, so that it would be easier for reader to understand the results findings and their derivation.

3) In structural modeling, the models have many amino acids (~5% in PURAwt and ~3.5% in PURA Phe233del) in disallowed region in Ramachandran plot, author(s) should describe why they have chosen these models, and also would have described that the number and type of amino acid residues are in disallowed region.

4)In Fig 6, author(s) should incorporate the distance in angstrom, and Figure 7 is also not clear, it would be better to show in different rotation, and avoid using heavy bold arrows.

5)Also author(s) would consider maintaining uniformity in figure depiction like in Fig 3 & 4 background is in white and in rest figures it's black, it would be better to keep one format. All the figures need to plot clearly.

6) Explain the QM-MM calculations in a clear way and also describe why you have performed that. Also decrease in area/volume and twisting effect is confusing. Author(s) should consider to address it in detail.

Author Response

Comments from Reviewer #2

  1. Authors might consider improving the quality of figures, and it still needs significant improvement, so that it would be clear to understand. 

The complete figures were redone following the points highlighted by Reviewer #2. Please see below.

  1. All the results headings look like methods headings e.g. “3.2. Protein Reconstruction “, author (s) would write the main finding highlights as one sentence headings of each result paragraph, so that it would be easier for reader to understand the results findings and their derivation.

Done. The headings of these paragraphs were amalgamated into the Results section text so that they are more accessible for the reader to understand.

  1. In structural modelling, the models have many amino acids (~5% in PURAwt and ~3.5% in PURA Phe233del) in disallowed region in Ramachandran plot, author (s) should describe why they have chosen these models, and also would have described that the for the number and type of amino acid residues are in disallowed region.

The model was chosen because for the AlphaFold prediction model, either for the PURAwt or PURA Phe233del, the best-obtained values maximize energetic level, protein size, and stereochemistry with a lower percentage in the disallowed regions. In the case of PURAwt, the number of residues in the disallowed region was 13, corresponding to Proline and Glycine residues harbored in the loops of the secondary structure; on the other hand, for PURA Phe233del, the number of residues in the disallowed region was 9, also constituted by Proline and Glycine residues.

This paragraph was added to the Results section.

  1. In Fig 6, author (s) should incorporate the distance in angstrom, and Figure 7 is also not clear, it would be better to show in different rotation, and avoid using heavy bold arrows.

We appreciate the reviewer’s comments and suggestions.

Regarding Figure 6, we now include the distance in Å. Kindly see our response to comment #3 of Reviewer #1.

As for Figure 7, we experimented with HDOCK and used different rotations following the reviewer’s suggestion. However, we concluded that the rotation and view presented in the manuscript best suited our proposal.

  1. Also author(s) would consider maintaining uniformity in figure depiction like in Fig 3 & 4 background is in white and in rest figures it’s black, it would be better to keep one format. All the figures need to plot clearly.

All figures in the revised manuscript were changed following the reviewer’s advice.

  1. Explain the QM-MM calculations in a clear way and also describe why you have performed that. Also decrease in area/volume and twisting effect is confusing. Author(s) should consider to address it in detail.

The QM-MM simulation allowed us to analyze the structural changes of PURAwt and PURAdel233Phe by dissecting at atomic resolution the electronic structure restricted to the targeted region where the mutation occurred. We used a semiempirical theoretical approach to the macromolecular level with an Austin Model1, which allows us to measure properties such as i) angle change and ii) surface potential; the rest of the PURA protein structural changes, due to computational costs, are described with classical physics from the MMFF basis. The changes in area/volume and angular twists imply steric changes that affect the functionality of the protein and its interaction; therefore, twists or reductions in area and volume space mean an interaction impairment with molecules such as nucleic acids, which is diminished with the deletion of an electronically bulky group such as the Phe233.

This paragraph was added to the discussion section.

Reviewer 3 Report

The manuscript “Analysis of the Structural Protein Effects caused by the PURA p.Phe233del Mutation Associated to Cognitive Developmental Delay using Artificial Intelligence and Hybrid Quantum Mechanics-Molecular Mechanics Modelling” by Lopez-Rivera et al present a first report of a de novo mutation underpinning a PURA syndrome in a Latin American patient. They also predict the molecular effects in protein structure using artificial intelligence algorithms.

Author Response

The manuscript “Analysis of the Structural Protein Effects caused by the PURA p.Phe233del Mutation Associated to Cognitive Developmental Delay using Artificial Intelligence and Hybrid Quantum Mechanics-Molecular Mechanics Modelling” by Lopez-Rivera et al present a first report of a de novo mutation underpinning a PURA syndrome in a Latin American patient. They also predict the molecular effects in protein structure using artificial intelligence algorithms.

Thank you for cleverly summarizing our work!

  1. Al-Numan HH, Jan RM, Al-Saud N bint S, Rashidi OM, Alrayes NM, Alsufyani HA, et al. Exome Sequencing Identifies the Extremely Rare ITGAV and FN1 Variants in Early Onset Inflammatory Bowel Disease Patients. Frontiers in Pediatrics [Internet]. 2022;10. Available from: https://www.frontiersin.org/article/10.3389/fped.2022.895074
  2. Buel GR, Walters KJ. Can AlphaFold2 predict the impact of missense mutations on structure? Vol. 29, Nature Structural and Molecular Biology. 2022.

Round 2

Reviewer 2 Report

Author (s) confused with the results section headings, and have removed it. However, in suggestion no. 2) All the results headings look like methods headings e.g. "3.2. Protein Reconstruction ", author(s) would write the main finding highlights as one sentence headings of each result paragraph, so that it would be easier for reader to understand the results findings and their derivation. 

This suggestion indicates that write the results heading in one sentence that reflect the findings of that paragraph, like "3.2.1 WES bioinformatics" "it should be something like that "3.2.1 WES bioinformatics indicates heterozygous mutation which affects functional domain of protein", and then results paragraph description. It would help to maintain the coherence of the results and their findings.

Figure 3, 4 and 6 still need to be improved.

Author Response

We want to express that we very much appreciated the excellent advice of Reviewer #2. With Her/his advice, we brought together a new dimension and collaboration within Universities, and our new manuscript is a much differentiated and significantly improved version of the original paper we wanted to publish. We hope that our manuscript accomplishes the last requirements of Reviewer #2.

Author (s) confused with the results section headings, and have removed it. However, in suggestion no. 2) All the results headings look like methods headings e.g. "3.2. Protein Reconstruction ", author(s) would write the main finding highlights as one sentence headings of each result paragraph, so that it would be easier for reader to understand the results findings and their derivation.

This suggestion indicates that write the results heading in one sentence that reflect the findings of that paragraph, like "3.2.1 WES bioinformatics" "it should be something like that "3.2.1 WES bioinformatics indicates heterozygous mutation which affects functional domain of protein", and then results paragraph description. It would help to maintain the coherence of the results and their findings.

Done. Sorry by the confusion. We made these corrections in the result section.

Figure 3, 4 and 6 still need to be improved.

New versions of the figures are now included in the reviewed version

This manuscript is a resubmission of an earlier submission. The following is a list of the peer review reports and author responses from that submission.

Round 1

Reviewer 1 Report

Dear Editor,

I went through the article and reviewed it. The present article has proposed that PURA-Phe233del mutation affect the specific surrounding region and predict the molecular effects in protein structure using artificial intelligence algorithms.

But still, I have observed lacking information and extensive interpretation of results and require more experimental support.

Thank you,

Reviewer 2 Report

The paper by Juan Javier Lopez-Rivera et al. entitled "Structural Protein Effects caused by the PURA p.Phe233del Mutation Associated to Cognitive Developmental Delay by Artificial Intelligence Modelling" describes structural analysis of a disease-associated Phe233del mutation in Purine-rich element binding protein A (PURA) discovered by NGS of a patient with Cognitive Developmental Delay. Authors discuss structural changes in PURA due the deletion.

Generally speaking, the work is interesting and lies within the trend for use of bioinformatics and molecular modeling for personalized medicine, to understand diseases and seek novel treatment.  

However, there are some significant limitations at present, which require further analysis and interpretation. In essence, authors predicted two 3D-models and discuss their differences without providing plausible mechanistic explanation, at least on the level of expert hypothesis, of why the observed structural changes are important to a function. At present, this does not amount to a scientific result, in my opinion. 

Key criticism: 

1/ Throughout the manuscript, authors praise the Alphafold algorithm, as it was their original development. While AF and similar algorithms indeed amount to a revolution in computational structural biology, such repetitive celebration of a non-original method in the current paper is not appropriate. AF being highly interesting yet remains a predictive algorithm. Its benefits, problems and limitations were discussed in previous publications and benchmarkings. Since the current paper is using AF to make computational prediction, there is no need to praise the AF itself. Instead, authors should focus on carefully describing and interpreting their results taking into consideration its predictive nature.
E.g., "... highlight the importance of predicting the protein structure molecular effects with artificial intelligence algorithms" - the idea of the paper is interesting, but I do not see how results highlight importance of AI/AF, both being interesting and useful, and yet limited in their predictive accuracy, and nothing in the AI area has changed with the current manuscript.

2/ While I find the idea of the study interesting and state-of-the-art, the presentation of the key result seems trivial to me. You do not need Alphafold to predict that a removal of one residue from the backbone in Phe233del-variant would result in a significant local structural rearrangement. Authors describe the differences between two predicted models -  WT versus Phe233del - in detail, but what is clearly lacking is its functional evaluation. How exactly Phe233del influences the protein function? Without a mechanistic explanation, at least on the level of expert hypothesis, of why these changes are important, this does not amount to a scientific result, in my opinion. Authors could use docking to construct complexes of WT and Phe233del with DNA/RNA and discuss changes at the expert level - as one possible and fast response to this criticism.

Other major points:

3/ I completely misunderstood the first part of Results. Throughout the text, authors state that they use AF to model the structures of the wild-type and phe233-deletion proteins. How does 697_699del fit in? Where did it come from. Is it part of the wild-type? 

4/ In the first part of Results authors discuss the difference in rotamer orientation of Phe233 (e.g. Fig1, c and d) predicted by Alphafold. While AF seems to be accurate in predicting the backbone and the overall fold, its accuracy in side-chain prediction is yet a matter of debate. Thus, fine details of AF-predicted side-chain orientations should be treated with caution. Molecular dynamics of AF-predicted structures as staring models can be used to validate the rotamer orientations.

5/ Section "Conclusion" is absent from the manuscript and should be added to summarize the study, its implications and limitations. 

6/ In respect to previous point, authors should clarify the novelty of their work. It was not clear to me whether the Phe233del mutation was previously known or discovered first time in this work.

Reviewer 3 Report

In this manuscript author’s presented idea that protein structure alteration occurred by PURA p.Phe233del mutation, however the following concerns of the manuscript are

  1. It lacks the clear presentation of figures, thereby it would be difficult for readers to understand.
  2. Protein models are used in this manuscript is not validated and it should be evaluated using SAVES or Molprobity, to check the stereochemistry.
  3. It should include more results figures/tables of sequencing quality control/methods or findings and genes with clear depiction.
  4. Results and discussion sections should be elaborated.

Reviewer 4 Report

The manuscript “Structural Protein Effects caused by the PURA p.Phe233del Mutation Associated to Cognitive Developmental Delay by Artificial Intelligence Modelling” is a communication article focused on the identification of the pathogenic mutation c.697_699del p.Phe233del of the PURA gene in an 11 y/o girl with cognitive developmental delay and on the investigation of the structural effects induced by this Phe233-deletion on the transcriptional activator protein Pur-alpha (PURA). According to the protein structural prediction by Alphafold, the deletion of Phe233 induces a distortion of the surrounding area, generating a disruption of the repeat III and an alteration of the β-sheet holding this residue. These structural effects drastically compromise the protein function. As communication article, this manuscript includes significant information to proceed further in the characterization of the effects induced by this gene mutation that leads to cognitive developmental delay. I would recommend manuscript publication after addressing the issues reported below.

Minor issues:

  1. Abstract, lines 14-19. The sentence is too long and not clear, please modify.
  2. Abstract, lines 24-25. “the Phe233del affects between 220-320 amino acids” should be changed to “the Phe233del alters the conformation of the region including amino acids 220-320”.
  3. Abstract, lines 25-27. The sentence is not clear, please modify.
  4. Section 1, lines 37-41. The sentence is too long and not clear, please modify.
  5. Section 1, lines 44-45. The sentence could be supported by a figure reporting the sequence alignment and highlighting the highly conserved repeats I-III.   
  6. Section 1, line 69. Since the structural analysis is based on a prediction model the term “demonstrate” seems inappropriate. “We demonstrate that after the p.Phe233del mutation,” could be modified to “In the predicted model of PURA Phe233del,”.
  7. “Phe 233” should be corrected to “Phe233” throughout the manuscript.
  8. Section 2, line 97. Please correct “1.0ug”.
  9. Figure 1. Panel a, the quality of the panel should be improved to make it more readable and the font size should be enlarged. Panel c and d, amino acid labels should be added to the figure.
  10.  Section 3. Before describing the mutated area, the authors should add a description of the whole structure, reporting the main functional elements, as the conserved repeats I-III.
  11. Section 3, lines 216-217. “the Phe233del affects between 220-320 amino acids” should be changed to “the Phe233del alters the conformation of the region including amino acids 220-320”.
  12. Section 3, lines 217-223. The sentence is not clear, please modify.
  13. Figure 2. An additional panel showing the structural superimposition of the predicted models of PURA and PURA-Phe233del should be added to the figure. It is difficult to appreciate comparative structural changes from panels c and d, showing a surface rendering of the proteins.

Reviewer 5 Report

The manuscript “Structural Protein Effects caused by the PURA p.Phe233del Mutation Associated to Cognitive Developmental Delay by Artificial Intelligence Modelling” by Lopez-Rivera et al present a first report of a de novo mutation underpinning a PURA syndrome in a Latin American patient. They also predict the molecular effects in protein structure using artificial intelligence algorithms. Accordingly, this reviewer recommends publication.